# Classification of Aggressive Movements Using Smartwatches

**DOI:** 10.3390/s20216377

**Published:** 2020-11-09

**Authors:** Franck Tchuente, Natalie Baddour, Edward D. Lemaire

**Affiliations:** 1Department of Mechanical Engineering, University of Ottawa, Ottawa, ON K1N 6N5, Canada; ftchu094@uottawa.ca (F.T.); nbaddour@uottawa.ca (N.B.); 2The Ottawa Hospital Research Institute, Ottawa, ON K1H 8M2, Canada; 3Faculty of Medicine, University of Ottawa, Ottawa, ON K1H 8M5, Canada

**Keywords:** aggressive movements, smartwatches, feature selection, machine learning classifiers, performance metrics

## Abstract

Recognizing aggressive movements is a challenging task in human activity recognition. Wearable smartwatch technology with machine learning may be a viable approach for human aggressive behavior classification. This research identified a viable classification model and feature selector (CM-FS) combination for separating aggressive from non-aggressive movements using smartwatch data and determined if only one smartwatch is sufficient for this task. A ranking method was used to select relevant CM-FS models across accuracy, sensitivity, specificity, precision, F-score, and Matthews correlation coefficient (MCC). The Waikato environment for knowledge analysis (WEKA) was used to run 6 machine learning classifiers (random forest, k-nearest neighbors (kNN), multilayer perceptron neural network (MP), support vector machine, naïve Bayes, decision tree) coupled with three feature selectors (ReliefF, InfoGain, Correlation). Microsoft Band 2 accelerometer and gyroscope data were collected during an activity circuit that included aggressive (punching, shoving, slapping, shaking) and non-aggressive (clapping hands, waving, handshaking, opening/closing a door, typing on a keyboard) tasks. A combination of kNN and ReliefF was the best CM-FS model for separating aggressive actions from non-aggressive actions, with 99.6% accuracy, 98.4% sensitivity, 99.8% specificity, 98.9% precision, 0.987 F-score, and 0.984 MCC. kNN and random forest classifiers, combined with any of the feature selectors, generated the top models. Models with naïve Bayes or support vector machines had poor performance for sensitivity, F-score, and MCC. Wearing the smartwatch on the dominant wrist produced the best single-watch results. The kNN and ReliefF combination demonstrated that this smartwatch-based approach is a viable solution for identifying aggressive behavior. This wrist-based wearable sensor approach could be used by care providers in settings where people suffer from dementia or mental health disorders, where random aggressive behaviors often occur.

## 1. Introduction

Dementia is a mental disorder that affects more than 35 million people in the world, and is expected to double over the next 20 years [1]. In long-term residential care, more than forty percent of the elderly are affected by this disorder. People with dementia can become quickly agitated, and verbally and even physically aggressive [2]. Kicking, hitting, or pushing are some aggressive behaviors frequently observed and listed in conventional scales such as the Cohen–Mansfield agitation inventory [3]. Direct observation by caregivers is the typical method for defining a person’s challenging behaviors [4]. However, observation is subjective, prone to diagnosis errors, reliant on caregivers being present for aggressive incidents, and might require excessive caregiver time. The ability to quickly recognize aggressive situations could result in prompt intervention and a better understanding of the person’s behavior. Tailored care could then be adopted to better deal with this problem and help both caregivers and patients.

Most movement aggressiveness research uses computer vision methods that rely on external systems such as optical sensors (cameras) that capture images or videos for classification. Vision-based equipment includes infrared cameras, depth cameras, RGB cameras, 3D motion capture sensors, Microsoft Kinect, or Vicon cameras [5]. These tools might be useful in sensitive environments with constant video-monitoring, such as prisons or psychiatric centers that are prone to fights, agitation, aggressiveness, and violence. 

Ouanane [6] recognized aggressive human behavior with two visual methods: Bag of features and skeleton graph. The recognition rate was 96% for activities such as boxing, hand clapping, hand waving, jogging, running, and walking. Deniz [7] identified extreme acceleration patterns to determine fights and violent sequences from surveillance videos, with 90% classification accuracy. Mecocci [8] proposed a maximum warping energy (MWE) approach to detect violent acts (fighting, falling) from normal behavior (walking, running, handshaking). MWE describes the spatial-temporal complexity of color conformation from video sequences and values were significantly greater for aggressive actions. While effective, image and video processing are computationally intensive and raise privacy issues, especially in clinical establishments or nursing homes where residents do not want to be continuously video recorded. 

Human activity recognition (HAR) technology could also be used to identify aggressive behavior. Wrist devices such as smartwatches have become popular, especially in the fitness and well-being industries. These devices use sensors and inertial data to provide daily health monitoring information (e.g., number of steps, calories, heart rate) [9]. Several studies have used smartwatches to classify writing, eating, sitting, and jumping, with accuracies ranging from 80% to 90%. Applications included a wrist-worn Actigraph to evaluate activity recognition and fall detection [10], a smartwatch system to identify gestures associated with writing the alphabet (94% to 99% accuracy) [11], and a Sony SWR50 smartwatch system that detected stereotyped movements in children with a development disability (i.e., clenching a fist, waving a hand, swinging an arm, raising an arm, lowering an arm, throwing) [12]. While research has addressed HAR activities such as walking, standing, or climbing stairs, research is lacking on aggressive movement recognition with wearable sensors.

Wrist-mounted inertial measurement units, such as a smart identification bracelet or smartwatch, have great potential for broad application within healthcare and elderly-care facilities and provide a repeatable location for capturing upper-limb related aggressive activities. Identification of aggressive events could improve service delivery by enabling an alarm-based notification of event onset, and could also provide quantitative information on who initiated the aggressive event, which is often difficult to understand in elderly care environments where dementia is prevalent. 

The goal of this research was to determine if smartwatches can effectively differentiate between aggressive and non-aggressive movements by determining a viable classification model and feature selector (CM-FS) combination using smartwatch inertial sensor data. A secondary objective was to determine the best location for the wrist-worn sensors system: bilateral wrists (BW), dominant wrist (DW), or non-dominant wrist (NDW).

The research presented in this manuscript has its foundation anchored on the original thesis [13].

## 2. Methods

### 2.1. Data Collection and Equipment

A convenience sample of 30 able-bodied people (15 male, 15 female) were recruited from the Ottawa Hospital Rehabilitation Centre (TOHRC) staff, students, volunteers, and the community. Characteristics included age (25.9 ± 8.0), weight (70.2 ± 11.9 kg), height (170.7 ± 8.6 cm), and right-handedness (20 out of 30). Thirty participants provided sufficient data for model training and evaluation in this research. The study was approved by the Ottawa Health Science Network and the University of Ottawa Research Ethics Boards. All participants provided informed consent.

Participants wore one Microsoft band 2 (MSB2) smartwatch per wrist and donned a holster on their pelvis that carried a Nexus 5 smartphone. The MSB2 recorded upper-limb motion via integrated tri-axial accelerometer and gyroscope sensors. The Nexus 5 smartphone was connected via Bluetooth to the smartwatches using the TOHRC data logger [14] Android app, updated for signal acquisition from two MSB2. A second smartphone video recorded participant movements and served as a gold standard comparator. The gold standard time was synchronized with the smartwatch sensor output by shaking the hands at the beginning and end of the trial, thus providing a recognizable accelerometer signal and video event. MSB2 tri-axial accelerometer and tri-axial gyroscope data (Figure 1a) provided linear acceleration and angular velocity. Having one watch per wrist: (i) provided movement data from both upper limbs, thus ensuring that all wrist motions were captured; (ii) enabled analysis to evaluate handedness; and (iii) enabled analysis to determine if data from one or both wrists would be sufficient for classification.

### 2.2. Activity Circuit

Participants performed an activity circuit that included non-aggressive and aggressive actions (Table 1). Similar activities, such as a slap and clapping, were chosen to present opportunities for misclassification. Aggressive actions were performed on a body opponent bag (BOB) (Figure 1b), a realistic humanlike and height adjustable combat dummy.

Aggressive and non-aggressive movements in general refer to a wider range of movements. However, for the purpose of this experiment, the aggressive and non-aggressive movements referenced in this manuscript mainly refer to the activities presented in Table 1.

### 2.3. Preprocessing and Feature Selection

Raw data were extracted from MSB2 accelerometers and gyroscopes sensors at 50 Hz (Figure 2, Figure 3 and Figure 4) for the complete activity circuit. Data were synchronized to the video at the beginning and the end of the activity. Sensor data were subsequently divided into 1 s sliding windows (50 data points) for feature extraction. One second was sufficient to classify quick aggressive activities [15]. 

The sliding window W had n points where $W\;=\;\left\{x_{1},\;x_{2},\;x_{3},\;\ldots ,\;x_{n}\right\}$ represents a 1-s time interval [$t_{a},\;t_{b}$]. The window label was a function of the window’s last data point $\left(x_{n}\right)$, obtained from the gold-standard video. 

While a 50% window overlap was used empirically [16], a 96% window overlap (window advanced by 2 data points) was used in this study to provide immediate activity analysis. Pre-analysis with window overlaps, varying between 50% and 98%, led to 96% as an appropriate window overlap. Time series data were converted into discrete variables for feature selection and extraction (Figure 5). This resulted in 122,307 training instances over the entire activity circuit. Data were not filtered prior to processing since the sliding window technique worked appropriately and incorporated data smoothing.

Selecting high-quality features can lead to better classification accuracy and decreased error rates. In this research, 68 time-domain features (Table 2) were initially chosen to classify aggressive and non-aggressive movements (136 features with both wrists). Time-domain features were chosen because they are less computationally expensive than frequency-domain features [10]. Fifty-six features were statistical and twelve features were based on movement (physical features). Three feature selection methods were used to select the 20 best features along each axis. Farah [17] found that classification accuracy improvements stopped increasing beyond selecting the 20 best features. These features were subsequently fed into the machine learning classifiers. Details of the features are given in Appendix A.

Information gain (InfoGain: IG) feature selection was implemented using WEKA (Waikato environment for knowledge analysis). InfoGain is a single-feature evaluator that measures the feature’s total entropy with respect to the class [12], employing a ranked search to provide a specific rank to each feature. The selected features were compared to evaluate dispersion around the class. ReliefF (ReF) is an instance-based evaluator that randomly samples instances and checks instances near the same and different classes. ReliefF has been heavily used in HAR studies [18]. Correlation (C) evaluates the worth of a feature by measuring Pearson’s correlations between that feature and the class, whereas the Chi-squared test evaluates features by computing the feature’s Chi-square statistics with respect to the class [19].

Six machine learning classifiers have been used extensively across HAR areas [20]: Random forests (RF), k-nearest neighbors (kNN), multilayer perceptron neural network (MP), support vector machines (SVM), naïve Bayes (NB), and decision trees (DT). WEKA [20] was used for all classifications. These machine learning classifiers were fed by feature sets from ReF, IG, and C. Aggressive or non-aggressive classification performance was determined for each combination of classification method and feature selector (CM-FS) using accuracy, sensitivity, sensibility, precision, F-score, and Matthews correlation coefficient (MCC). For example, RF-ReF refers to the random forest classifier fed by the relief-F feature set.

A summed ranking classifier selection method was used to combine several metrics to rank CM-FS performance [21]. The summed ranking method ranks classification models in descending order (best results ranked as 1) according to each metric. The ranks for all six metrics were subsequently summed to provide an overall ranking for each model. Models were sorted in descending order, since the lowest rank value indicated the best model. This ranking method gave a better, wider, and more generalizable representation of model performance since results from all six parameters were considered.

## 3. Results

### 3.1. Bilateral Smartwatches Classification

For sensor data from two smartwatches (bilateral smartwatches or BW), the best model combination was kNN ReF, with 99.6% accuracy, 98.4% sensitivity, 99.8% specificity, 98.9% precision, 0.987 F-score, and 0.984 MCC (Table 3). The kNN-ReF confusion matrix is shown in Table 4. kNN-ReF, RF-IG, kNN-C, kNN-IG, RF-C, and RF-ReF were the top 6 models, with average metrics of 99.16% accuracy, 95.75% sensitivity, 99.77% specificity, 98.82% precision, 0.9715 F-score, and 0.9670 MCC.

### 3.2. Unilateral Smartwatch Classification

Using two smartwatches provided high classification metrics. However, smartwatches are generally worn on one wrist. Therefore, two additional scenarios were considered: Sensor data from the dominant wrist (DW) and sensor data from the non-dominant wrist (NDW). The top model for bilateral classification (kNN-ReF) was used to evaluate the three scenarios (Table 5). BW had the best results: 99.6% accuracy, 98.4% sensitivity, 99.8% specificity, 98.9% precision, 0.987 F-score, and 0.984 MCC. DW and NDW had comparable results, with a slight advantage towards DW.

## 4. Discussion

### 4.1. Bilateral Smartwatches Classification

This research demonstrated that aggressive and non-aggressive motions can be classified using accelerometer and gyroscope data from smartwatches. kNN-ReF was the best combination for this binary classification. These results were comparative or better than computer vision approaches that scored between 91% and 96% [6]. Therefore, the proposed smartwatch method represents a viable way of identifying aggressive movements, possibly leading to a wearable system for alerting care providers to an aggressive event and logging information to better understand the aggressive situation.

kNN-ReF ranked first across all performance metrics, except precision where kNN-ReF was ranked second. All kNN-ReF performance metrics were above 98.4%. Even though ReliefF was the worst ranked feature selector in general, it worked well with kNN. One explanation is that ReliefF uses inherently nearest neighbors to estimate attribute relevance; therefore, ReliefF would be compatible with the kNN machine learning classifier. Villacampa [19] also noticed that combining ReliefF and kNN improved results in a binary classification. kNN-ReF was followed by RF-IG, kNN-C, kNN-IG, and RF-C in the group of five best models. Performance measures for these combinations were consistently in the top 5.

The worst model was NB-ReF, which had acceptable accuracy (85.46%) and specificity (91.10%) but low sensitivity (54%), precision (52%), F-score (0.53), and MCC (0.44). This would result in a high number of false positives (aggressive actions incorrectly classified as non-aggressive) and false negatives (non-aggressive actions incorrectly classified as aggressive). The five lowest ranking models were NB-ReF, NB-IG, MP-ReF, NB-C, and SVM-IG. SVM-ReF had especially low sensitivity (15.20%), F-score (0.2610) and MCC (0.346). Given these results, the naïve Bayes and SVM classifiers should not be used as machine learning tools to recognize aggressive movement using inertial sensor features.

For individual performance measures, sensitivity was high for kNN-ReF, kNN-C, and kNN-IG. High sensitivity indicates few false negatives (aggressive actions not detected by the classifier), so these models are ideal if the priority is to identify all aggressive events. Precision represents false alarms (actions classified as non-aggressive that, in reality, are aggressive). SVM-ReF, NB-ReF, and MP-ReF were the best ranked precision models. Therefore, these models are suitable if the main criterion is to minimize false positives. F-score combines sensitivity and precision, but does not consider the correctly identified non-aggressive actions. MCC is a balanced measure that takes into account all the four confusion matrix components and is very useful when there is a class imbalance. F-score and MCC ranking results were the same and displayed similarities with the general summed ranking results. kNN-ReF, kNN-C, and kNN-IG were the top 3 F-score and MCC models.

In this research, modeling and machine learning analyses were performed offline, meaning that the results were not obtained from a real-time system (i.e., a device that instantly notifies staff when an aggressive event occurs). Applying the selected models in a real-time platform might yield different and probably lower performance metrics [22].

Furthermore, ten activities were considered in the groups of aggressive and non-aggressive movements. It would be interesting to evaluate the impact of a wider range of movements on the observed metrics.

### 4.2. Unilateral Smartwatch Classification

Both unilateral and bilateral smartwatch approaches were effective, with excellent results. Since all results were above 92%, regardless of selected wrist, any approach could be used in a clinical care setting and achieve satisfactory results. However, the best results occurred with bilateral smartwatches, where BW had better results than DW and NDW across the six metrics. When comparing BW and DW, the main differences appeared in specificity (false positives, where non-aggressive situations might not be detected by the algorithm). DW and NDW results were similar, with differences less than 1%, in favor of DW, across the metrics (Table 5).

Specificity was greater than 0.931 for all smartwatch conditions. This is important when considering implementation in an assistive or nursing care facility since false alarms would have a negative impact on work flow, and care staffing levels are often minimal.

If achieving the highest performance metrics is a priority, smartwatches should be worn on both wrists. However, using two watches might be cumbersome and introduce added expense. The minor outcome differences between two watches (BW) and one watch (NDW or DW) could support the unilateral condition due to cost and convenience factors for the user and the care team. Since DW had a slight advantage over NDW in terms of outcome measures, DW would be the preferred configuration. With the objective to minimize false positives, DW leads with 98.1% accuracy, 98.1% sensitivity, 93.6% specificity, 98.1% precision, 0.981 F-score, and 0.926 MCC, which are very reasonable results for a binary classification.

## 5. Conclusions

A smartwatch-based approach for identifying aggressive activity was investigated to determine a viable classification model and feature selector (CM-FS) combination for separating aggressive from non-aggressive movements. The kNN classifier and ReliefF feature selection combination provided excellent aggressive movement classification results, with all performance metrics above 98%. Using this model, an alarm-based notification of aggressive events would lead to a miss rate of only 1.6% (incorrectly classifying an aggressive action as non-aggressive) and a false alarm rate of less than 0.2% (incorrectly classifying a non-aggressive activity as aggressive). The metrics support use of this model in a clinical setting to identify aggressive events by means of smartwatches or other wrist-worn devices (e.g., inertial sensor ID bracelet). Other models such as RF-IG or kNN-C are suggested if the focus is on minimizing false positives or false negatives.

Using one smartwatch per wrist provided excellent classification results, with performance metrics exceeding 92%. False positives and false negatives that can easily occur in machine learning classification were minimized. Despite the best performance results, adopting two smartwatches requires additional financial, computational, and practical resources. Therefore, one smartwatch on the dominant wrist can be considered when implementing an aggressive movement identification application.

Future research in this area should include model evaluation within a real-time system, testing with an elderly population that reflects people with dementia in healthcare settings, and a multinomial classification that attempts to distinguish each of the aggressive and non-aggressive movements.

## Figures and Tables

**Figure 1 sensors-20-06377-f001:**
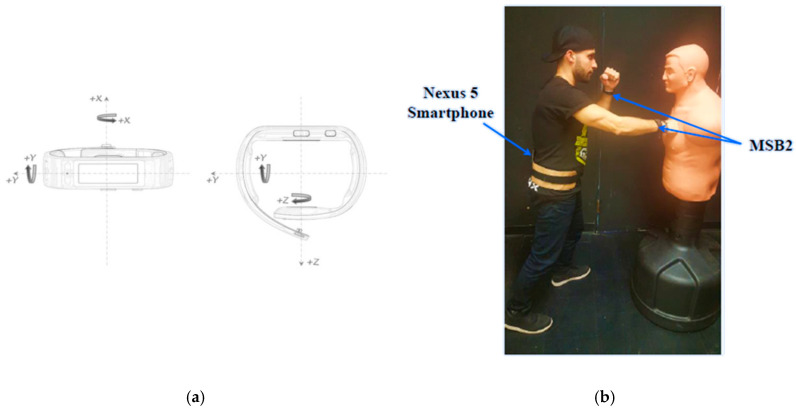
(**a**) Microsoft band 2 (MSB2) accelerometer and gyroscope axes orientation. (**b**) Participant punching the body opponent bag.

**Figure 2 sensors-20-06377-f002:**
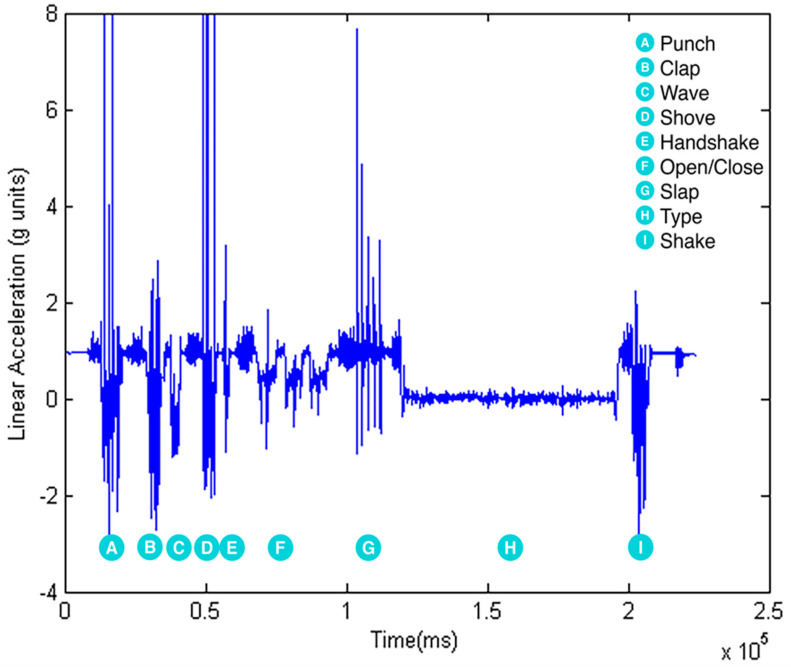
Accelerometer linear acceleration (x-axis).

**Figure 3 sensors-20-06377-f003:**
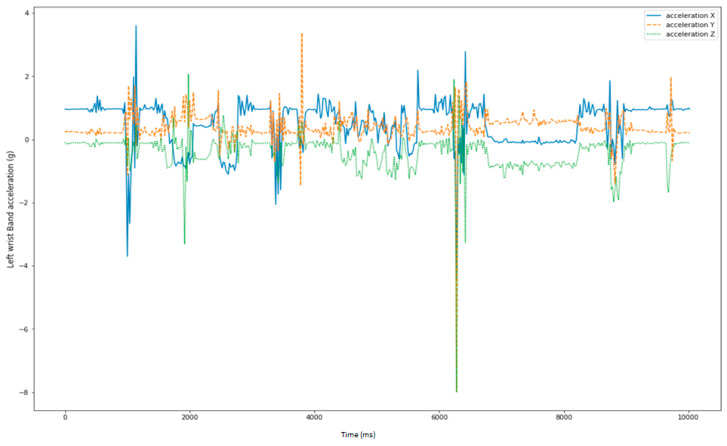
Tri-axial linear acceleration of participant 1.

**Figure 4 sensors-20-06377-f004:**
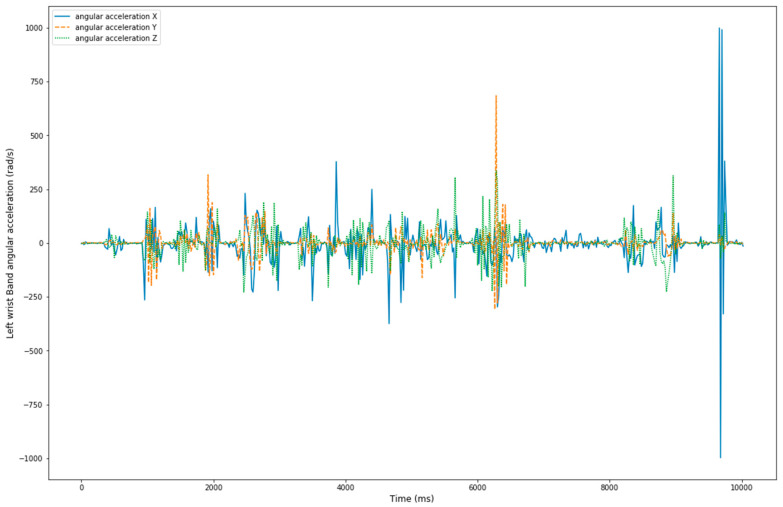
Tri-axial angular acceleration of participant 1.

**Figure 5 sensors-20-06377-f005:**
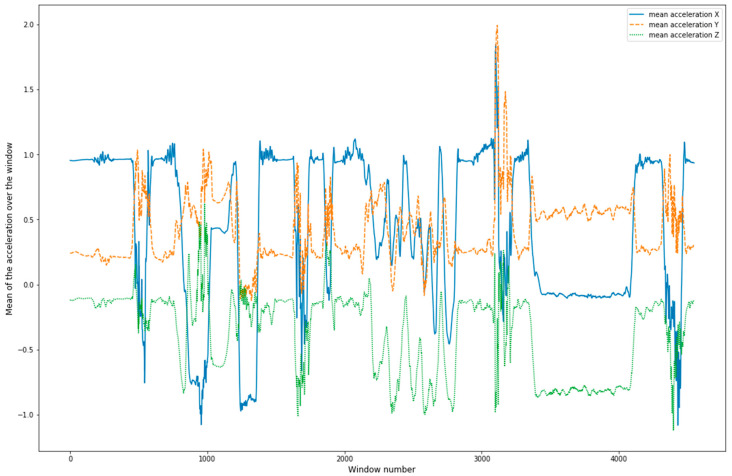
Extracting the mean feature from raw data sliding windows.

**Table 1 sensors-20-06377-t001:** Activities.

Movement	Activity	Description
Aggressive movements	Punch	Participant punches BOB eight times, alternating hands
	Shove	Participant aggressively shoves BOB five times with both hands
	Slap	Participant aggressively slaps BOB ten times, alternating hands
	Shake	Participant holds BOB’s neck and shakes BOB’s back and forth five times
	Transitions	Set of movements between an aggressive action and non-aggressive action (i.e., sitting, standing, moving, still)
Non-aggressive movements	Clap	Participant claps their hands ten times
	Wave	Participant waves with the preferred hand as if they are saying goodbye
	Handshake	Participant handshakes the project assistant
	Open/close door	Participant opens and closes the door three times
	Type on a keyboard	Participant types the first verse of the Canadian national anthem

**Table 2 sensors-20-06377-t002:** Feature description.

Feature	Description	# Features
**Statistical Features**		
**Mean**	Average of the signal	**6**
**Variance**	Variance of the signal	**6**
**Median**	Median of the signal	**6**
**Range**	Range of the signal	**6**
**Standard Deviation**	Deviation from the signal mean	**6**
**Skewness**	Asymmetry of the sensor signal distribution	**6**
**Kurtosis**	How peaked the sensor signal distribution is	**6**
**Pairwise Correlation Coefficient**	Correlation between two sensor axes, and between accelerometer and gyroscope sensors	**6**
**Integral**	Area under the curve	**6**
**Sum of All Squares**	Acceleration magnitude squared and summed over three axes $SaS\left(x_{i}\right)=\;\sum^{\;}a_{x}^{2}\left(x_{i}\right)+\sum^{\;}a_{y}^{2}\left(x_{i}\right)+\;\sum^{\;}a_{z}^{2}(x_{i}$)	**2**
**Physical Features**		
**Movement Intensity**	Average movement intensity (MI): The Euclidean norm of the total acceleration vector after removing the static gravitational acceleration, where $a_{x}\;\left(x_{i}\right),a_{y}\;\left(x_{i}\right)$, and $a_{z}\;\left(x_{i}\right)$ represent the $t_{th}$ acceleration sample of the x, y, and z axis in each window, respectively. $MI\left(x_{i}\right)=\sqrt{a_{x}^{2}\left(x_{i}\right)+a_{y}^{2}\left(x_{i}\right)+a_{z}^{2}\left(x_{i}\right)}$	**4**
**Signal Magnitude Area (SMA)**	The acceleration magnitude summed over three axes within each window normalized by the window length	**2**
**Maximum Difference**	Difference between the highest and the lowest value of over the window	**6**

**Table 3 sensors-20-06377-t003:** Classification method and feature selection combination sorted by summed rank (best to worst).

	**Score**					
	**Acc.**	**Sens.**	**Spec.**	**Prec.**	**FS**	**MCC**
kNN-ReF	0.996	0.984	0.998	0.989	0.987	0.984
RF-IG	0.992	0.962	0.998	0.998	0.974	0.970
kNN-C	0.995	0.979	0.997	0.985	0.982	0.979
kNN-IG	0.994	0.974	0.997	0.983	0.979	0.975
RF-C	0.990	0.949	0.998	0.986	0.967	0.962
RF-ReF	0.983	0.897	0.998	0.988	0.94	0.932
DT-IG	0.985	0.941	0.993	0.959	0.95	0.941
DT-C	0.983	0.932	0.992	0.956	0.944	0.934
MP-C	0.966	0.837	0.989	0.93	0.881	0.862
MP-IG	0.965	0.841	0.988	0.924	0.881	0.862
DT-ReF	0.959	0.844	0.98	0.883	0.863	0.839
SVM-C	0.949	0.774	0.98	0.876	0.822	0.794
SVM-ReF	0.870	0.152	0.998	0.931	0.261	0.346
SVM-IG	0.945	0.756	0.979	0.865	0.807	0.777
NB-C	0.935	0.822	0.955	0.767	0.793	0.756
MP-ReF	0.933	0.722	0.97	0.812	0.764	0.727
NB-IG	0.929	0.813	0.949	0.742	0.776	0.735
NB-ReF	0.855	0.54	0.911	0.52	0.53	0.444
	**Rank**					
	**Acc.**	**Sens.**	**Spec.**	**Prec.**	**FS**	**MCC**
kNN-ReF	1	1	1	2	1	1
RF-IG	4	4	1	1	4	4
kNN-C	2	2	6	5	2	2
kNN-IG	3	3	6	6	3	3
RF-C	5	5	1	4	5	5
RF-ReF	8	8	1	3	8	8
DT-IG	6	6	8	7	6	6
DT-C	7	7	9	8	7	7
MP-C	9	11	10	10	9	9
MP-IG	10	10	11	11	9	9
DT-ReF	11	9	12	12	11	11
SVM-C	12	14	12	13	12	12
SVM-ReF	17	18	1	9	18	18
SVM-IG	13	15	14	14	13	13
NB-C	14	12	16	16	14	14
MP-ReF	15	16	15	15	16	16
NB-IG	16	13	17	17	15	15
NB-ReF	18	17	18	18	17	17

Acc. = Accuracy, Sens. = Sensitivity, Spec. = Specificity, Prec. = Precision, FS = F-score, MCC = Matthews correlation coefficient.

**Table 4 sensors-20-06377-t004:** K-nearest neighbors (kNN)-ReliefF (ReF) confusion matrix.

		**True Condition**	
		Aggressive	Non-Aggressive
**Predicted Condition**	Aggressive	18,073	189
	Non-aggressive	306	103,739

**Table 5 sensors-20-06377-t005:** Performance metrics (kNN-ReF).

	Accuracy	Sensitivity	Specificity	Precision	F-Score	MCC
**BW**	0.996	0.984	0.998	0.989	0.987	0.984
**DW**	0.981	0.981	0.936	0.981	0.981	0.926
**NDW**	0.980	0.980	0.931	0.980	0.980	0.920

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
