# Peer review of "Classification of Aggressive Movements Using Smartwatches"

_sensors, 2020, doi:10.3390/s20216377_

Round 1

Reviewer 1 Report

see attachment

Reviewer 2 Report

The rationale for developing this tool for the dementia population is not well justified. Patients are not generally aggressive by themselves, only with interacting with someone. So, if someone is present, why can’t they just observe the behavior and report on it? Why do you need a smart watch?

The methodology is not quite sound. You took very able bodied people doing exaggerated movements and showed you can tell a difference using highly selected parts of the recording.

What happens when you take mostly bed bound, feeble dementia patients? Do you think there might be limitations of detection?

How were the time domain features selected for further analysis? What happens when you don’t select features?

Can this be used in real time?

How might it be used clinically?

Minor comments: describe or reference bag of features, skeleton graph, shaking the hands.

Round 2

Reviewer 2 Report

No further comments.